# It Takes More than Two to Tango: Complex, Hierarchal, and Membrane-Modulated Interactions in the Regulation of Receptor Tyrosine Kinases

**DOI:** 10.3390/cancers14040944

**Published:** 2022-02-14

**Authors:** Tamas Kovacs, Florina Zakany, Peter Nagy

**Affiliations:** Department of Biophysics and Cell Biology, Faculty of Medicine, University of Debrecen, 4032 Debrecen, Hungary; kovacs.tamas@med.unideb.hu (T.K.); florina.zakany@med.unideb.hu (F.Z.)

**Keywords:** receptor tyrosine kinases, protein–protein interactions, plasma membrane, fluorescence techniques, dipole potential

## Abstract

**Simple Summary:**

Receptor tyrosine kinases probably constitute the most important subfamily of transmembrane receptors with respect to their role in regulating the balance between cell proliferation and cell death. Their activation involves ligand-induced conformational changes followed by their dimerization. Although this simple mechanism is still known to lie at the root of the process, the picture is complicated by the involvement of several receptor domains in the dimerization and the formation of larger receptor aggregates. Both clustering and activation are influenced by lipid-mediated interactions of the plasma membrane with the receptors. The intricate regulation of receptor activation is subverted in cancer that involves not only alterations in receptor structure and expression but also changes in lipid composition of the cell membrane. This paper provides a concise overview of how these biophysical aspects of transmembrane signaling regulate this important process in health and disease.

**Abstract:**

The search for an understanding of how cell fate and motility are regulated is not a purely scientific undertaking, but it can also lead to rationally designed therapies against cancer. The discovery of tyrosine kinases about half a century ago, the subsequent characterization of certain transmembrane receptors harboring tyrosine kinase activity, and their connection to the development of human cancer ushered in a new age with the hope of finding a treatment for malignant diseases in the foreseeable future. However, painstaking efforts were required to uncover the principles of how these receptors with intrinsic tyrosine kinase activity are regulated. Developments in molecular and structural biology and biophysical approaches paved the way towards better understanding of these pathways. Discoveries in the past twenty years first resulted in the formulation of textbook dogmas, such as dimerization-driven receptor association, which were followed by fine-tuning the model. In this review, the role of molecular interactions taking place during the activation of receptor tyrosine kinases, with special attention to the epidermal growth factor receptor family, will be discussed. The fact that these receptors are anchored in the membrane provides ample opportunities for modulatory lipid–protein interactions that will be considered in detail in the second part of the manuscript. Although qualitative and quantitative alterations in lipids in cancer are not sufficient in their own right to drive the malignant transformation, they both contribute to tumor formation and also provide ways to treat cancer. The review will be concluded with a summary of these medical aspects of lipid–protein interactions.

## 1. Introduction

Receptor tyrosine kinases (RTK) play a central role in regulating the number of cells in tissues that determine the intricate balance between cell proliferation, cell survival, and cell death. This equilibrium is subverted in cancer, and activation of RTKs is often causally related to the development and spread of malignancy [1]. Physiological and pathological activation of RTKs involves clustering, which is regulated not only by protein–protein interactions but by the influence of the membrane environment as well, into which these growth factor receptors are embedded. In this review article, conformational rearrangements and concomitant changes in clustering will be described with the aim of showing how the latter regulate and fine-tune the former. Special attention will be given to protein–lipid alterations and their potential role in cancer. Since from among RTKs, the epidermal growth factor (EGF) receptor family is the most frequently involved in oncogenesis, the most attention will be paid to this subfamily.

## 2. Dimerization-Induced Activation of Receptor Tyrosine Kinases

RTKs are single-pass transmembrane proteins that transduce extracellular cues through the membrane. How a single alpha-helix can achieve this feat perplexed scientists for decades, but in the last twenty years structural biology made great progress towards answering this question. It has been shown that ligand binding induces conformational changes in the extracellular domain of these receptors and that this altered conformation is conducive to dimerization [2,3]. Resolution of the crystal structures of the liganded and unliganded forms of several such receptors established that, although this general principle applies to practically every case, several different mechanisms exist with regard to the details of protein–protein interactions [1]. In the first mode, receptor dimerization is mediated by the ligands without direct contact between the extracellular domain of receptors, as in the case of the TrkA receptor [4]. In the second kind of mechanism, dimeric ligands begin the process that also involves receptor–receptor interactions, e.g., in the case of the KIT receptor and the platelet-derived growth factor (PDGF) receptor [5,6]. In the third mode, interactions between the monomers are exclusively receptor mediated, as in the case of the epidermal growth factor receptor family that will be introduced in more detail later [3]. If there are constitutive, preformed dimers, ligand binding does not simply induce dimerization but changes the arrangement and conformation of these preformed, inactive receptor dimers. This can be brought about by a single ligand, as in the case of the growth hormone receptor [7], or by two ligands, e.g., the effect of insulin on its receptor [8,9]. Finally, there are examples for the involvement of accessory molecules in the process, as in the case of the activation of the fibroblast growth factor (FGF) receptor (FGFR) that involves a 2:2:2 FGF:FGFR:heparin ternary complex, in which heparin facilitates growth factor binding [10].

### 2.1. The ErbB Family of Receptor Tyrosine Kinases

The EGF receptor (EGFR) is unique among RTKs in many respects. It was the first such receptor to be discovered about half a century ago [11], it is one of the most widely studied [12], and it is the most important from the standpoint of cancer. Depending on the histological type of the tumor and the population studied, the prevalence of its mutation and overexpression can be as high as 30–40% in human malignancies [13]. It is the founding and eponymous member of the ErbB family, also called the EGFR family, and it is activated by a variety of peptide growth factors (EGF, transforming growth factor α (TGFα), amphiregulin, epigen, epiregulin, betacellulin, and heparin-binding EGF) [14]. The third and fourth members of the family, ErbB3 and ErbB4, also known as HER3 and HER4, bind to a family of ligands, heregulins, or neuregulins [15]. According to the dominant view, ErbB3 harbors an inactive pseudokinase lacking several key conserved and catalytically important residues for which it appears to be locked into an inactive conformation [16]. At the same time, several lines of evidence have been presented for ErbB3 preserving enough catalytic power to be capable of tyrosine phosphorylation, especially in the presence of ErbB2 [17,18]. ErbB4, on the other hand, is a fully competent receptor capable of neuregulin binding and tyrosine phosphorylation [19]. Still, it is made unique by the existence of several alternatively spliced variants that differ in their potential to undergo intramembrane proteolytic cleavage, followed by nuclear translocation, and to activate PI3K signaling [20,21]. ErbB2 is also unique in the family since it is assumed to function as a shared coreceptor for other ErbB proteins, enhancing their signaling potency and increasing their ligand affinity [22,23]. It has no known soluble ligand, but the EGF-like domain of the MUC4 sialomucin has been shown to activate it [24,25]. This interaction is also relevant from the standpoint of cancer therapy since the large size of the MUC4 complex, brought into proximity of ErbB2, results in masking of the ErbB2 epitope for trastuzumab, a monoclonal antibody used in the treatment of ErbB2-positive breast cancer [26].

### 2.2. Role of the Extracellular Domain of ErbB Receptors in Dimerization

In order to understand the mechanism of receptor activation, the domain structure of ErbB receptors must be described. All ErbB proteins comprise an extracellular, ligand-binding portion that is itself divided into four domains numbered I-IV. The extracellular domain (ECD) is connected by a single, alpha-helical transmembrane domain (TMD) and a juxtamembrane domain (JMD) to the tyrosine kinase domain (KD) that is followed by the C-terminal regulator domain harboring tyrosine residues phosphorylated upon receptor activation [27] (Figure 1A).

One of the greatest breakthroughs in the molecular understanding of how EGF receptor family members are activated came at the beginning of the new millennium when a series of publications resolved the mechanism of ligand-induced activation of this class of RTKs. The ECD of all three ligand-binding ErbB receptors was shown to adopt a closed or tethered conformation characterized by two important features: (i) The ligand-binding pocket for high-affinity growth factor binding does not exist since the receptor parts involved in ligand binding are spatially separated from each other; (ii) The dimerization arm involved in forming intermolecular connections in dimers is buried inside the receptor [28,29,30] (Figure 1B). While the intramolecular tether seemed essential for the autoinhibited conformation of the ECD based on these crystal structures, it was later shown that other receptor domains must contribute to the autoinhibited conformation as well [31,32]. It was also recognized that large-scale molecular rearrangements must take place in order to make high-affinity ligand binding and dimerization possible. Crystallographic studies of the EGFR in complex with its ligand revealed that the ligand stabilizes a substantially different, extended conformation of the ECD in which the dimerization arm is exposed, and the ligand-binding pocket is formed [33,34] (Figure 1B). The ligandless, orphan receptor of the family, ErbB2, was shown to adopt a conformation in which the dimerization arm is continuously exposed, grossly resembling liganded EGFR structures [35]. However, this structure of the ErbB2 ECD recapitulates several key features of the unliganded *Drosophila* EGFR, in which several autoinhibitory intramolecular interactions are present [36]. The crystal structure of ErbB2 also revealed electrostatic repulsion between the dimerization arm and the pocket into which it could dock in trans, suggesting that back-to-back homodimers, observed for liganded EGFRs, cannot form between ErbB2 molecules [35]. A different kind of homodimer exhibiting a back-to-head arrangement has been found in crystallographic studies of ErbB2 homodimers [37].

### 2.3. Involvement of the Transmembrane and Kinase Domains in Receptor Dimerization

While the previous paragraph presented a compelling picture linking structural rearrangements of the ECD to receptor activation, it is not only the ECD that undergoes substantial changes during activation. While in monomeric receptors, the conformations of the extra- and intracellular domains are likely to be uncoupled from each other [38], they are usually assumed to be correlated in dimeric structures. In the remainder of this section, we are going to describe what is known about the activation mechanisms of individual transmembrane and kinase domains and discuss how they cooperate with each other in the activation process.

The fact that a Val→Glu amino acid substitution in the TMD of ErbB2 is an oncogenic mutation showed a long time ago that the TMD fulfills an important role in regulating dimerization [39,40]. Three out of the four ErbB receptors, ErbB1, ErbB2, and ErbB4, harbor a conserved, GxxxG-like dimerization motif at both the N- and the C-terminal ends of their TMD, while this motif is present only at the N-terminus of the ErbB3 TMD [41]. Several lines of evidence support that the two terminal GxxxG motifs stabilize different kinds of TMD dimers. Dimerization mediated by the C-terminal motif, which is close to the intracellular face of the TMD, is present in inactive dimers, whereas dimerization through the N-terminal motif is involved in dimerization-dependent activation [39,42,43]. It has been suggested that the membrane dipole potential, to be discussed later in this review, stabilizes the association of the TMDs through the N-terminal dimerization motifs, thereby enhancing ErbB receptor-mediated signaling [44].

Although the basic tenet of growth factor-induced activation of the EGFR kinase has been known for a long time, the activation mechanism was a conundrum since phosphorylation of the KD, an essential step in the activation of most kinases, was found to be dispensable in EGFR [45,46]. In a groundbreaking study, it was revealed that the KD of one of the receptors in a dimer, the activator, plays a role analogous to that of a cyclin bound to a cyclin-dependent kinase and activates the kinase of the second receptor, the receiver, in an asymmetric dimer [47,48]. Besides this asymmetric dimer, the KDs can dimerize in a symmetric fashion; however, the kinases in this arrangement are unlikely to be capable of signal transduction [49,50]. The same principle of kinase activation has been proposed to take place in EGFR/ErbB2 heterodimers as well [51]. Biophysical studies also allowed insight into the mechanism of action of EGFR-directed tyrosine kinase inhibitors. The two major classes of these small molecule inhibitors differ in whether they recognize the active or inactive conformation of the KD [52]. Several studies found that type I inhibitors, stabilizing the active conformation of the kinase, increase the formation of EGFR dimers harboring inhibited KDs, while type II inhibitors, stabilizing the inactive kinase structure, inhibit homodimerization altogether [53,54,55]. However, lapatinib, a type II inhibitor, has been shown to latch ErbB2 into a conformation that predisposes it to form head-to-head KD dimers with ErbB3, which explains the unexpected synergy between this kinase inhibitor and heregulin in promoting cell proliferation [56].

### 2.4. Coupling between Different Receptor Domains in the Dimerization Process

In the previous section, the involvement of distinct receptor domains was outlined, ending with the activation mechanism of the KD. According to the emerging picture, the inherent tendency of the KD to form active dimers has to be kept under control by the concerted action of the extracellular, transmembrane and juxtamembrane domains. Ligandless dimers of the ECD have been shown to prevent the formation of active, asymmetric KD dimers [57]. According to molecular dynamics (MD) simulations, the ECD dimerizes even in the absence of ligand, and although the ECD is in a “partially extended” conformation, the structure of the dimer is different from that of EGF-bound ECD dimers. Such ligandless dimers hold the N-terminal dimerization motifs of the TMD apart, favoring TMD dimerization through their C-terminal dimerization motif, binding of the JMD to anionic phospholipids, and the formation of symmetric, inactive kinase dimers [58]. On the contrary, liganded dimers of EGFR ECDs favor dimerization of TMDs through their N-terminal dimerization motifs, dissociation of the juxtamembrane segment from the membrane, and the formation of active, asymmetric KD dimers. The principle of coupling TMD dimerization through the N-terminal dimerization motif to dissociation of the JMD from the membrane has also been observed for ErbB2 [59].

The complex interactions between receptor domains are also involved in ligand discrimination, i.e., the activation of distinct signaling outcomes depending on which particular ligand activates the same receptor. Although both EGF and TGFα stabilize grossly similar ECD dimers, the conformation of the TMD dimers, and consequently, the arrangement of the JMDs significantly differ [60]. High- and low-affinity EGFR ligands differ even more profoundly since epigen, a low-affinity EGFR ligand, stabilizes an asymmetric ECD dimer different from the “canonical”, EGF- and TGFα-stabilized dimer. This feature is manifested in different dimer stability and longevity [61], which in turn results in distinct tyrosine phosphorylation patterns in the C-terminus [62]. These experiments also show that interpretation of dynamic properties of receptor interactions, which can only be studied in live cells, are indispensable for an accurate description of receptor activation, laying the foundation for the next section of the review.

There is disagreement about the cooperativity of ligand binding to EGFR. Since cooperativity is influenced by the conformational changes induced by a liganded receptor in its unliganded counterpart, it sheds light on the dynamic changes taking place after growth factor binding. Several studies reported negative cooperative EGF binding curves [63,64], and these studies have been rationalized by linking negative cooperativity to the apparent heterogeneity of EGF binding sites [65], and by the observation of a squeezed, restrained ligand-binding site in the unliganded receptor in a singly liganded dimer in the *Drosophila* EGFR [66]. However, several findings put the issue into a different perspective, including the significant structural differences between how human and *Drosophila* EGFRs are activated [36], the loss of negative cooperative binding in studies of isolated, human EGFR ECDs [67], and the repeated observation of positive, cooperative ligand binding to EGFR [53,68,69,70]. It has been proposed that the inactive and active conformations of the KD are coupled to the tethered and untethered structures of the ECD, respectively, assumed to display different cooperativities [70]. A comprehensive, quantitative model [53] invoked the formation of chains of preformed, unliganded dimers harboring symmetric kinase dimers [71] and their escape from these superclusters to explain positive cooperativity. This model also accounts for the expression level dependence of apparent cooperativity and for the existence of preformed dimers [53].

### 2.5. Biophysical Studies Reveal the Complexity of Clustering

Although the conformation of receptors and their dimerization is essential for keeping unstimulated receptors in control and making their ligand-induced activation possible, several lines of evidence suggest that the formation of clusters larger than dimers and their dynamic properties also contribute to the physiological regulation of signaling. Such associations are typically not amenable to structural biological techniques but can be investigated by fluorescence-based biophysical approaches. While superresolution techniques, enabling surpassing the resolution limit of conventional light microscopy, have lately come of age [72], older approaches, such as fluorescence correlation spectroscopy (FCS) [73], fluorescence recovery after photobleaching (FRAP) [74], single particle tracking (SPT) [75,76], and Förster resonance energy transfer (FRET) [77] supplemented by MD simulation [78], have their legitimate role in the research of membrane proteins. They offer insight into temporal and spatial dimensions not easily reachable by superresolution approaches [79]. In this section, a brief review of such studies on ErbB receptor clustering is provided.

While structural data, detailed in the previous sections, point at the existence of ligand-independent dimers harboring inactive KD dimers, the observation of such preformed, constitutive dimers was difficult to reconcile with the widely held view of ligand-induced dimerization. Therefore, their existence and relevance have been debated ever since they were proposed [80]. Preformed dimers have been repeatedly observed using FCS, especially close to the cell periphery [81,82]. While high EGFR expression levels were found to favor their formation [82,83], the concentration of cholesterol in the membrane inversely correlated with their abundance [84]. Similar to structural studies, biophysical experiments involving tracking of single quantum dots also revealed that preformed dimers are different from liganded dimers. Ligand-independent dimers and dimers with only one of the receptors binding EGF are more transient and less stable than bona fide, double-liganded dimers. Furthermore, the comigration of two EGFRs was even observed in cases when their distance precluded direct molecular interactions. The authors concluded that such “co-confinement” was necessary to account for the observed transient nature of many dimerization events that were interrupted by short breaks followed by the same two receptors dimerizing again [85].

FRET measurements not only allow insight into clustering but also provide information about molecular conformations [77], a feature that has been applied to the EGFR as well. By measuring the distance between domain I of EGFR and the membrane, the conformational change involving the extension of the receptor away from the membrane upon ligand binding was observed in living cells [86]. FRET measurements between fluorescently labeled EGFs led to the conclusion that ligand-activated dimers must assemble to larger tetrameric structures in order to account for the smaller distance between two fluorescently labeled EGFs derived from FRET measurements, compared with the predicted distance between growth factors in the back-to-back crystallographic dimers [87]. Multimeric complexes of the EGFR have been repeatedly observed both in the absence and presence of stimulation [88,89,90]. Based on a combination of imaging and simulation techniques, it has been proposed that ligandless EGFRs form a chain of oligomers in which extended ECDs form head-to-head interactions, and the KDs are prevented from interacting with each other. According to this study, constitutive activation of EGFR is prevented by distancing of the KDs [71], a conclusion that is at odds with previous findings suggesting that symmetric, inactive KD dimers are responsible for the same [58]. Ligand binding induces conformational changes in these preformed dimers that have been suggested to involve rotation of the TMD [91], tetramerization [88,92,93], or the formation of even larger oligomers stabilized by ligand-free ECDs. While the KD itself can be active in a dimer, phosphorylation of C-terminal tyrosine residues required for transmembrane signaling was proposed to be possible in the context of the tetramer model only by means of transdimer phosphorylation [93]. The requirement for larger receptor oligomers for efficient signaling is supported by other lines of evidence, including the correlation between MAPK activation and the abundance of clusters with more than three monomers, and the correlation between large clusters and signaling efficiency, as suggested by total internal reflection and superresolution microscopy [94,95]. Different aspects of the aforementioned model involving different kinds of EGFR multimers before and after ligand stimulation have been supported by independent experimental evidence. EGF stimulation has been suggested to dissociate EGFRs from preformed clusters [96]. Constitutive, ligand-independent structures of EGFR have been shown to be stabilized by interactions of the JMD with anionic phospholipids [97], while EGF-induced oligomerization and subsequent signaling was reduced by mutations in domain IV of the extracellular region [98].

Biophysical studies allowed important insight into the workings of other ErbB proteins as well. According to homo-FRET measurements, nonactivated ErbB2 is sequestered in large clusters involving tens of receptors from which growth factor-activated ErbB1 or ErbB3 recruit ErbB2 for heterodimerization [89]. This model is supported by the increased lateral mobility of ErbB2 after heregulin stimulation [99]. While the amount of experimental data available for ErbB3 is comparatively small, the emerging overall picture is quite similar. Ligand independent oligomers of ErbB3 are disassembled by its ligand, heregulin [100,101]. Similar to EGFR, ErbB3 also cycles repeatedly through homodimeric and heterodimeric states, with ErbB2 even in the absence of ligand, and heregulin binding stabilizes these dimeric structures [102]. Single-molecule tracking demonstrated that ErbB3 is largely monomeric before stimulation when expressed as the only member of the ErbB family, but the abundance of homodimers increased significantly by ErbB2 coexpression. Heregulin stimulation induced the competitive formation of homodimers of ErbB3 and ErbB2/3 heterodimers. Dimeric structures differed significantly concerning their mobility, with ligand-independent ErbB3 homodimers moving significantly faster than heregulin-induced ErbB3 homodimers or ErbB2/3 heterodimers [103].

Experimental evidence accumulated during the past decades points at hierarchal clusters of dimers and higher-order oligomers, and both layers of associations undergo rearrangement upon receptor activation. Besides these two hierarchical levels of clustering, both ErbB1 and ErbB2 have been shown to form a third level of associations with characteristic diameters in the hundreds of nanometer range, which are beyond the scope of the current review [104,105]. This three-level domain organization seems to be a general feature of membrane proteins [106,107]. While there are some inconsistencies between the potential interpretation of results discussed in the previous paragraphs, a plausible model is presented in Figure 2. Contradictions are likely to stem from inherent differences in the methods and experimental conditions. X-ray diffraction investigates static crystals, and its scope is currently limited to the examination of receptor domains instead of whole transmembrane proteins. While most fluorescence-based biophysical methods look at receptors in intact cells, there are tremendous differences in terms of their requirement for fixation, potentially generating fixation artifacts, their temporal and spatial resolution, and the potential influence of labels on the behavior of receptors [79]. The detection of receptor oligomers by many authors and the complete lack of such multimeric structures in the observations of others is likely to be explained by such methodological differences.

## 3. Protein–Lipid Interactions and Their Examination

The previous section provided a structural overview of the transition between the inactive and active states of the EGFR from the standpoint of the protein itself. Involvement of the plasma membrane in the process has already been implied by the dimerization of the TMD, but a range of other protein–lipid interactions take part in controlling receptor activation. This section of the review is devoted to the importance of lipid-mediated interactions in the modulation of RTKs and to methods used for investigating these interactions.

Besides acting solely as passive diffusion barriers, biological membranes are essential active components for a wide variety of cellular functions, ranging from compartmentalization, leading to distinct microenvironments in the interior of cells, to information processing for signaling pathways. The great abundance of functionally relevant membrane-coupled cellular phenomena is facilitated by mutual interactions between lipids and peripheral and integral proteins, which can substantially influence their structure and function. For example, lipids can modulate associations of proteins through binding at the interface, influence the stability of conformational states, thereby shifting their equilibrium, fine-tune binding efficiency of ligands or other interacting partners, etc. In general, modulation of proteins by membrane lipids is thought to occur via two mechanisms, i.e., through specific, direct interactions or indirect effects mediated by alterations in membrane biophysical parameters, including fluidity, hydration, elastic compression, bending rigidity and dipole potential. While such arbitrary discrimination between direct and indirect lipid-mediated effects can be useful from a didactic point of view, it should be kept in mind that under physiological circumstances, their combination occurs in most cases [108,109]. In addition, proteins can induce local or even large-scale perturbations in the membrane, resulting from hydrophobic mismatch and electrostatic interactions, accompanied by the formation of a unique “lipid fingerprint”, a microenvironment specific to the given protein giving rise to a complex and nonuniform perturbation of local and dynamic membrane properties [110].

The examination of direct interactions between membrane proteins and lipids is challenging due to serious technical limitations and, therefore, most commonly carried out in a nonphysiological environment. For example, lipids strongly bond to proteins and thus surviving crystallization and isolation can be identified with X-ray crystallography, cryo-electron microscopy, or native mass spectrometry [111,112,113]. Weaker short-lived dynamic associations between proteins and lipids and accompanying conformational changes can be analyzed using hydrogen–deuterium exchange mass spectrometry, in which mass increase resulting from the exchange can be mapped at different locations in molecular complexes to visualize structural dynamics after extraction of proteins. Therefore, results do not necessarily reflect the native structure of membrane complexes [114,115]. Besides these experimental techniques, due to improvements in computing power and theoretical models, all-atomic and coarse-grained MD simulations became suitable to predict and explain transient protein–lipid interactions with good spatial and temporal resolution [108,116,117]. While these methods are invaluable for the examination of protein–lipid interactions, their detailed description is beyond the scope of the current review.

### 3.1. Examination of Direct Lipid–Protein Interactions Using Fluorescence Methods

Recent advances in fluorescence-based techniques paved the way for the examination of direct protein–lipid interactions in living cells with high temporal and spatial resolution. Molecular associations can reliably be identified using various FRET-based approaches after labeling the interacting partners with a donor and an acceptor fluorophore. When the two molecules are in close proximity to each other, i.e., the distance between them is <10 nm, a FRET signal can be detected, which can be manifested in sensitized acceptor emission, donor quenching, donor and acceptor photobleaching, or in altered fluorescence anisotropy, which can quantify molecular associations including direct protein–lipid interactions [118,119]. In these experiments, intrinsic amino acid residues of proteins, such as Trp located in TMDs, can act as donors, while fluorescently labeled lipids can act as acceptors. In an early study, decreased FRET efficiencies between Trp residues of nicotinic acetylcholine receptors (nAchR) and the fatty-acid derivative 6-dodecanoyl-2-dimethylaminonaphthalene (Laurdan) acting as donor and acceptor, respectively, in the presence of cholesterol or dioleoyl-phosphatidylcholine (DOPC) suggested the presence of direct lipid-binding sites in the protein [120]. Similar experiments in bovine retinal rod outer segment membranes demonstrated decreased donor quenching of Trp in rhodopsin molecules by cholestatrienol, a fluorescent sterol, in response to cholesterol incorporation proposing direct binding of the lipid to the protein [121]. Spectrofluorometric measurements in a liposomal FRET assay revealed direct and highly specific association between a fluorescently-labeled pentanoyl analog of C18-sphingomyelin (acceptor) and the TMD of the COPI vesicular transport machinery protein p24 (its Trp residue serving as donor) and identified VXXTLXXIY as a specific sphingolipid-binding signature sequence in mammalian membrane proteins [122]. In other cases, proteins are labeled with a fluorophore for FRET experiments. For example, when examining CD3ε using a fluorescent reporter protein attached to the C-terminus of the protein serving as FRET donor and R18 (a lipophilic octadecyl rhodamine B derivative incorporated into the membrane) serving as FRET acceptor, quantification of donor quenching with fluorescence microscopy demonstrated incorporation of the CD3ε cytoplasmic domain into the inner leaflet of the plasma membrane mediated by clusters of basic residues interacting with anionic phosphatidylserine (PS) and phosphatidylinositol species [123]. FRET measurements can also be combined with fluorescence lifetime imaging microscopy (FRET-FLIM) since energy transfer results in a decrease in the fluorescence lifetime of the donor, which is independent of changes in fluorescence intensity, i.e., the density of target molecules, a phenomenon advantageous when determining lipid–protein associations [118,124]. FRET-FLIM experiments in time-correlated single-photon counting (TCSPC) mode applying FRET donors and acceptors mentioned above revealed that TCR engagement of peptide–MHC complexes induced the release of the cytoplasmic domain of CD3ε from the plasma membrane, which might facilitate recruitment of signaling molecules [125].

Besides FRET, other fluorescence- and single-molecule-based techniques can also provide information about molecular interactions. For example, measuring the fluorescence fluctuations of a fluorophore with FCS or that of two fluorescently labeled molecules, such as a protein and a lipid, and determining their correlation in fluorescence cross-correlation spectroscopy (FCCS) can reveal diffusion characteristics of the particles revealing their interactions [126]. Since conventional FCS and FCCS techniques are typically applied in a confocal microscope, the relatively low resolution could complicate the interpretation of results suggesting direct associations. This limitation can be overcome by combining these methods with superresolution techniques, such as stimulated emission depletion (STED) microscopy providing a lateral resolution of <50 nm [127]. STED-FCS was used to demonstrate transient trapping of Atto670N-labeled sphingomyelin (SM), GM1 ganglioside, or GPI-anchor (but not phosphatidylethanolamine) in cholesterol-mediated molecular complexes within <20 nm areas in the membrane of living mammalian cells [128]. Similar STED-FCS experiments carried out in plasma membranes of living cells, using different lipid species tagged with lipophilic Atto647N with a focal spot size of 30 nm, gave further details about the transient formation of molecular complexes. Determination of anomalous diffusion revealed weak interactions for the examined phosphoglycerolipids, stronger cholesterol-assisted and cytoskeleton-dependent interactions for sphingolipids, and similarly hindered diffusion for gangliosides and galactosylceramides, which was much less cholesterol- and actin-dependent. These results suggested transient binding of these lipids to immobile or slowly moving membrane components, and while the exact identity of the partner was not determined, membrane-associated proteins were speculated as interacting molecules [129]. While conventional STED-FCS detects signals from a single small spot at a time, scanning STED-FCS (sSTED-FCS) allows rapid recording of FCS data along a line or a circle providing the basis for a more accurate spatiotemporal characterization of the lipid interactions by direct mapping of apparent diffusion coefficients of fluorescent analogs with appropriate spatial and temporal resolution. This technique was utilized to show that diffusion speed and trapping characteristics of Atto647N-labeled SM strongly change in space and time across the plasma membrane with the occurrence of transient hotspots for molecular interactions, presumably with proteins [130]. Similarly, when examining the mesoscale organization of proteins with metabolic incorporation of modified amino acid L-azidohomoalanine, followed by labeling with KK114 fluorophore conjugated to aza-dibenzocyclooctyne (DIBAC) using click reaction in living cells under physiological conditions and membrane sheets, STED and sSTED-FCS demonstrated the presence of long-lived multiprotein complexes characterized by restricted lateral diffusion, whose formation largely depended on cholesterol, and, to a lesser extent, the actin cytoskeleton. Specific proteins were found to show a preferential pattern of localization in different regions of the areas from their cores to their edges, suggesting the importance of specific lipid–protein interactions for the segregation process [131]. Subsequent STED-FCS studies emphasized the role of the actin cortex since hindered diffusion of most examined fluorescently labeled lipids (saturated phosphatidylethanolamine (PE), SM, but not GM1 ganglioside with the latter being involved in actin-independent, transient nanodomains) observed in living cells disappeared in cell-derived, actin cytoskeleton-free giant plasma membrane vesicles. Similar results were found when examining several GPI-anchored proteins, as these molecules were organized in large, static clusters and highly mobile transient nanodomain pools, and both were missing in vesicles [132]. Further details about the accurate membrane-organizing principles with a more detailed and direct elucidation of lipid–protein interactions might be revealed in the future using dual-color STED-FCCS.

Transient trapping of membrane molecules, and consequently their nanoscale interactions, can be analyzed by optical, single-fluorescent-molecule tracking methods other than STED, which can also provide the appropriate spatial (<30 nm) and temporal (<1 ms) resolution by following the two-dimensional trajectories of lipids or proteins labeled with small fluorescent dyes in living cell membranes [133]. Such single-fluorescent-molecule imaging and tracking measurements carried out with a TIRF microscope in membranes of living cells provided the first direct evidence for the interaction between GPI-anchored receptors and gangliosides since they revealed extremely dynamic, transient (10–40 ms) colocalization and codiffusion of fluorescently labeled GM1 and GM3 ganglioside analogs and fluorescently labeled CD59, a raft-resident, GPI-anchored protein in a cholesterol- and GPI-anchor-dependent manner [134]. Using this technique, similar, short-lived (12–50 ms), cholesterol- and GPI-anchor-dependent colocalization and codiffusion with CD59 was found when investigating novel fluorescent SM analogs [135].

### 3.2. Examination of Indirect Lipid-Mediated Effects on Proteins Using Fluorescent Methods

Besides direct interactions, lipid-mediated effects on the structure and function of proteins can also occur via alterations in bulk membrane biophysical properties, such as membrane fluidity, hydration, and dipole potential. Fluidity and hydration are two strongly related order parameters of biological membranes, with the former defined as a restriction of rotational freedom of membrane-incorporated molecules, and the latter as the extent of membrane penetration of water molecules. On the other hand, the dipole potential arises due to the preferential, nonrandom alignment of molecular dipoles of membrane lipids and water molecules localized at the bilayer–water interface resulting in an immensely strong, intramembrane electric field. While the mechanisms of how these properties affect proteins are scarcely described in detail, there are several proposed models for their actions, including hydrophobic mismatch, elastic coupling theories, and electrostatic interactions between charged regions of proteins and ordered membrane-associated dipoles, which can lead to altered stabilities of certain protein conformations or modified association-clustering tendencies, reviewed in [109,136,137,138,139,140].

These membrane biophysical parameters can be examined using a variety of methods, including NMR and ESR spectroscopy, MD simulations, wide-angle X-ray scattering, cryo-electron microscopy, and atomic force microscopy [141,142,143,144,145,146,147,148]. However, these techniques are not suitable for the investigation of living cells, which can rather be examined with membrane-incorporating, environment-sensitive fluorophores that change their excitation or emission characteristics in response to alterations of a local biophysical parameter. For example, the fluidity of biological membranes can be tested with TMA-DPH (4′-(trimethylammonio)-diphenylhexatriene) using spectrofluorometry, since its fluorescence anisotropy negatively correlates with the rotational freedom of the fluorophore, i.e., fluidity of the membrane, as shown in living cells with altered levels of glucosylceramide, cholesterol, other sterols, and saturated or polyunsaturated fatty acids [149,150,151,152,153]. Membrane hydration can be estimated using Laurdan (6-dodecanoyl-N,N-dimethyl-2-naphthylamine) since the value of generalized polarization quantifying shifts in its emission spectrum shows an inverse correlation with the degree of water penetration into bilayers [154,155]. Changes in membrane hydration in response to various lipids were demonstrated in living cells using spectrofluorometry [149,150,151]. While TMA-DPH and Laurdan are useful for spectrofluorometric approaches examining bulk solutions, their applicability for methods providing information about individual cells is limited by their excitability falling in the UV range. This can be overcome by two-photon microscopy [150,156], or, alternatively, using PY3174 (4-[2-(6-Dibutylamino-5-fluoro-naphthalen-2-yl)-vinyl]-1-(3-triethylammonio-propyl)-pyridinium dibromide), a Laurdan analogue with more convenient spectral properties [157]. The generalized polarization of PY3174, just as that of Laurdan, also shows a negative correlation with membrane hydration in living cells, which can be examined by conventional confocal microscopy. Therefore, signals can be collected exclusively from the cell membrane, eliminating the contribution of internalized dye molecules [151,153]. Investigation of dipole potential in living cells is mainly carried out with di-8-ANEPPS (4-(2-[6-(dioctylamino)-2-naphthalenyl]ethenyl)-1-(3-sulfopropyl pyridinium inner salt), a voltage-sensitive dye based on electrochromism, i.e., shifts in the excitation and emission spectra in response to changes in the magnitude of the surrounding, intramembrane electric field. This dye is most reliably used in excitation ratiometric assays when the ratio of fluorescence intensities integrated between two excitation wavelength ranges positively correlates with the magnitude of dipole potential in spectrofluorometry [149,153,158,159] or fluorescence microscopy [44,160]. Alternatively, F66 (N-[3-(40-dihexylamino-3-hydroxy-flavonyl-6-oxy)-propyl] N,N-dimethyl-N-(3-sulfopropyl)-ammonium inner salt), a 3-hydroxyflavone derivative can be applied instead of di-8-ANEPPS via an emission ratiometric assay due to its electric field-modulated, excited-state intramolecular proton transfer (ESIPT) reaction resulting in normal and tautomer excited states with well-separated bands in its emission spectrum. After excitation at a single wavelength, the ratio of its fluorescence intensities corresponding to the normal and tautomeric forms negatively correlates with the magnitude of the dipole potential in spectrofluorometric and fluorescence microscopic applications [160,161]. Furthermore, unlike di-8-ANEPPS [162], F66 was recently shown to be suitable for flow cytometric determination of the dipole potential, which allows high-throughput examination of this enigmatic biophysical parameter in large quantities of individual living cells [153].

### 3.3. Modulation of Receptor Tyrosine Kinases by Membrane Lipids

#### 3.3.1. General Considerations of Lipid Effects on Receptor Tyrosine Kinases and the Role of the Transmembrane Domain

Consistent with their single-pass TMDs, RTKs in general and ErbB proteins, in particular, were found to be largely affected by different lipid components of the cell membrane, as summarized in Figure 3. As described in Section 2.2 and Section 2.3, various domains of ErbB receptors undergo large conformational changes during activation, and several steps of the process occur in connection with the membrane. For example, in the resting monomer or inactive dimer states, the tethered conformation of the ECD ensures that the N-termini of the transmembrane helices of monomers are held apart. Thus, TMDs are consequently monomers or dimers associated via the C-terminal GxxxG dimerization motifs [39,163]. This configuration facilitates the embedding of the JM-A, the N-terminal segment of the JMD, into the membrane via the interaction between its positively charged residues and negatively charged lipids of the inner leaflet of the cell membrane, which stabilizes the inactive monomeric or symmetric dimeric state of the KD [58,164]. On the other hand, the open conformation of the ECD permits dimerization of TMDs through the N-terminal dimerization GxxxG motif [39,163], which enables the release of JM-A from the membrane. Consequently, extensive, antiparallel dimerization between JMDs of monomers can occur, leading to the formation of the catalytically active, asymmetric kinase dimer [47,58,164]. Besides these canonical conformations, other ECD, TMD, and JMD dimers or higher-order oligomers can form in the membrane, which can also fine-tune the activation of ligand type-dependent signaling mechanisms [56,60,93,98,163,165,166]. Further emphasizing the mutual connection with the membrane, the presence of ErbB receptors may affect the local concentration of lipid species as suggested by mass spectrometry of EGFR-containing lipid rafts and MD simulations of EGFR localized in a more realistic bilayer environment [110,167].

As described above, the TMD has an essential contribution to the activation of ErbB proteins. MD simulations of ErbB TM helices in various bilayer models demonstrated that a certain local lipid environment could contribute to the selection of a preferred conformation by changing the relative stabilities of different homo- and heteromeric ErbB TMD associations mediated through surface landscape complementarity and hydrophobic matching between lipids and the proteins, and the lipid environment can therefore, modulate the activation of downstream signaling events [58,163,168,169]. While other RTKs are characterized by slightly different activation mechanisms, lipid modulation of these processes follows similar principles. In general, RTKs usually have various TMD dimer conformations, and bilayer-dependent selection of a preferred TMD configuration occurs in membranes of different compositions, which essentially contributes to the modulation of RTK activation, as described recently for other members of this superfamily as well [170,171,172,173].

#### 3.3.2. Lipid-Mediated Effects on the JMD of RTKs with a Special Emphasis on Phospholipids

Besides the TMD, the JMD also plays an active role in RTK regulation. MD simulations of TMD–JMD constructs and their NMR analysis revealed correlations between the conformational changes of these domains [163,164]. Examination of nearly full-length receptors revealed the importance of membrane embedding of the JM-A segment as a bilayer composition-dependent electrostatic mechanism for receptor autoinhibition [58]. Anionic phospholipids of the inner leaflet can participate in direct electrostatic interactions with three positively charged amino acid clusters of the JM-A segment of ErbB proteins, as was demonstrated in MD simulations, leading to membrane burial of the segment. These interactions were energetically more favorable for inactive than active receptor configurations; therefore, they were suggested to be responsible for maintaining the inactive states of EGFR [58]. In accordance with this, solid-state NMR and fluorescence spectroscopy using TMD–JMD peptides of Neu (rat ErbB2) confirmed tight interactions of JMD with negatively charged lipids, and activating V664E mutation in the TMD leads to the release of the JMD from the membrane surface, which is associated with JMD dimerization [59]. Besides PS species, other negatively charged membrane lipids can act as interaction partners for basic residues of ErbB receptors. Surface plasmon resonance of a peptide mimicking the EGFR JMD revealed strong binding to phosphatidylinositol-4,5-bisphosphate (PIP_2_), and in cellular studies downregulation of PIP_2_ levels or neutralization of negatively charged amino acids in the membrane-proximal, charged cluster of the JMD abolished this interaction, which was accompanied by reduced EGF-induced EGFR autophosphorylation and signaling [174]. Consistently, a fluorescence polarization assay showed binding of EGFR JMD to bicelles containing 10% PIP_2_, which was much stronger than binding to those with 50% PS despite more negative charges in the latter suggesting specific binding. The ionic JMD-PIP_2_ interaction was confirmed in living cells using a FRET-based approach. Furthermore, STORM revealed PIP_2_ clustering overlapping with EGFR clustering in membranes of lung cancer cells and normal lung epithelial cells, and EGFR clustering and downstream signaling were reduced after PIP_2_ depletion or mutation of membrane-proximal EGFR residues [97]. Subsequent multiscale MD simulations of EGFR TMD–JMD corroborated JMD-PIP_2_ binding by demonstrating PIP_2_ clustering near a group of basic amino acids at the start of the JMD (R645-R647 representing the first of the three charged amino acid clusters), aiding the stabilization of JM-A dimer away from the membrane and the consequent formation of the active asymmetric kinase dimer. Mutation of these residues or lower PIP_2_ levels led to decreased contact and consequent alterations in JM-A dimer structure [175]. Free energy landscape analysis using coarse-grained simulations confirmed the stable and energetically favorable interaction between PIP_2_ and EGFR JMD and the substantial role of R645-R647 in determining the lipid selectivity of the interaction [176]. Single-pair FRET imaging with TIRF of fluorophore-conjugated EGFR TMD–JMD peptides introduced into nanodiscs confirmed the PIP_2_-facilitated dimerization of JM-A segments, which was abolished in response to phosphorylation of Thr654, a target site for inhibitory phosphorylation in the JM-A, suggesting the regulation of electrostatic JMD–PIP_2_ interaction by this residue [177]. Furthermore, this electrostatic interaction was recently hypothesized to be influenced by the magnitude of the membrane dipole potential. Namely, increased magnitude of this intramembrane positive potential was found to facilitate ErbB2 dimerization and autophosphorylation, which was proposed to occur due to increased, dipole potential-mediated repulsion of basic residues in the JM-A segment by the positive lobe of the dipole potential, enhancing the formation of the active JMD–KD configuration [44]. Supporting the presence of a conserved interaction pattern involving binding of JMDs to anionic lipids among other members of this receptor superfamily, multiscale MD simulations suggested that JMDs of all human RTKs induce local bilayer reorganization and clustering of anionic lipids including PIP_2_ and PS, which is mainly mediated by a conserved cluster of basic residues within the first five positions of the JM region and by negatively charged headgroups of lipids in the inner leaflet. These measurements also proposed that while N-terminal JMD residues are involved in specific PIP_2_ binding, distal amino acids show much lower lipid specificities. Such interactions can be functionally relevant by modulating the nanoscale organization and functional activity of receptors [178]. Consistently, lipid-dependent, JMD-mediated regulation of receptor function was described for other RTKs such as FGFR3 [179], EphA2 receptors [180,181] and TrkA [172,182].

#### 3.3.3. Effects of Cholesterol on RTKs

Consistent with the preferential localization of ErbB proteins in cholesterol-enriched lipid raft microdomains of the cell membrane, cholesterol was shown to substantially affect the function of these receptors at various functional levels. Cholesterol is generally thought to inhibit the functional activity of ErbB receptors, since its experimentally reduced cellular levels enhanced ligand binding [183,184,185], subsequent receptor association [82,84,184,186], and autophosphorylation [84,183,184,186,187], and activation of downstream signaling such as MAP kinase activation [183,186,188]. In keeping, cholesterol replenishment of the cell membrane generally reverted these alterations [84,183,184,185,186,187]. These effects were generally attributed to indirect actions of cholesterol, such as alterations in the raft partitioning and consequent changes in the receptor microenvironment characterized by physical properties such as decreased fluidity and the presence of potential interacting partners leading to the release of the receptors from inhibitory constraints [82,84,183,187,188]. Emphasizing the mutual interaction between membrane biophysical parameters and ErbB proteins, overexpression of these receptors was shown to result in bilayer deformation [189] and cholesterol-dependent changes in local thickness or curvature of the membrane were proposed to alter their clustering [93]. In seeming contradiction with those described above, an additional level of complexity was added to lipid raft-mediated effects on ErbB function by studies demonstrating upregulation of cholesterol biosynthesis and elevated cholesterol levels in lipid rafts of human lung and breast cancer cell lines treated with EGFR tyrosine kinase inhibitors (TKI), which correlated with resistance to this treatment. Furthermore, pharmacological depletion of cholesterol levels by lovastatin or ketoconazole led to reduced EGFR signaling and sensitization of these cell lines to EGFR TKI both in vitro and in vivo, which was partially attributed to lipid raft disruption [190,191,192]. Similarly, lovastatin potentiated the effects of lapatinib, an ErbB2 TKI, to strongly suppress in vitro and in vivo growth of ErbB2 positive breast cancer xenografts [193].

Besides these supramolecular actions, cholesterol can also influence ErbB proteins through direct interactions. As shown by MD simulations, cholesterol molecules accumulate at the GxxxG dimerization motifs of the TMD regions suggesting its role in dimerization. Furthermore, cholesterol was proposed to favor the formation of the functionally active N-terminal TMD dimerization, which was attributed to its effects on membrane thickness [194]. This might be due to its possible contribution to the rigid, “frozen” pad of lipids described previously at the C-terminus of TMD, facilitating TMD dimerization through the N-terminal motif and consequent receptor activation [195]. Cholesterol-facilitated, activating dimerization is also supported by previous findings demonstrating that increases in the magnitude of the cholesterol-dependent dipole potential might facilitate TMD dimerization through N-terminal GxxxG over the C-terminal one by decreasing repulsion between helical dipoles in the former configuration [44]. These seemingly contradictory, supramolecular and direct cholesterol effects on ErbB functions are further underlined by findings suggesting the exit of these receptors from lipid rafts during activation [44].

Membrane cholesterol was shown to modulate functions of other RTKs at both supramolecular and molecular levels as well. Unlike in the case of ErbB proteins, cholesterol depletion and consequent disruption of caveolar structures or lipid rafts typically result in attenuated signaling activity. This was shown for insulin receptor-induced signaling in adipocytes [196] and hepatocytes [197], insulin-like growth factor 1 receptor activation that can be restored by various sterols [198], aldosterone-induced PDGFR- and Src-mediated proinflammatory signaling in vascular smooth muscle endothelial cells [199], NGF-elicited TrkA-mediated pathways modulating differentiation [200], and BDNF-induced signaling through TrkB in synaptic modulation and plasticity of neurons [201]. Recent findings emphasized the direct action of cholesterol on RTKs, and particularly TrkB. Cholesterol supplementation of primary cortical neurons enhanced BDNF-induced TrkB phosphorylation and its interaction with the downstream effector phospholipase C-γ1 and consequent neurite branching, which were reverted by cholesterol lowering. MD simulations revealed that alterations in cholesterol levels could change the orientation of TMD dimers between signaling competent and incompetent configurations mediated by GxxxG motifs in a manner similar to EGFR [202]. This was proposed to be mediated by direct binding, as TrkB proteins and other RTKs such as insulin receptor, FGFR1, FGFR4, Ephrin-type A receptor, and ErbB4, contain cholesterol-recognition amino acid consensus (CRAC) motif and its inverted version (CARC). MD simulations confirmed that cholesterol can affect TrkB through binding to such a motif [203]. Interestingly, a recent study applying magic angle spinning NMR spectroscopy examining the V664E mutant form of Neu suggested a putative CARC motif at the N-terminus of the TMD, which does not map to a single helical face, but the residues appear on opposite helical faces. According to the proposed highly speculative model, a pair of helices could create the motif, and cholesterol binding to this region might disrupt active N-terminal TMD dimers. However, other configurations not detectable with the applied probes could maintain both protein–protein and protein–cholesterol interactions [204]. Coarse-grained simulations of ErbB2 TMD-JMD peptides embedded into various bilayers showed prominent cholesterol accumulation around the TMDs, especially at the described CRAC motif overlapping the C-terminal GxxxG motif, and the cholesterol interactions were shown to substantially modulate the dimer energy landscape favoring dimeric forms over monomers, with the presence of JMD providing further structural stabilization. It was hypothesized that competition between the C- and N-terminal protein–protein and protein–cholesterol interactions determine the resulting conformational landscape of the several possible configurations, leading to a dynamic piston-like motion of TM helices. However, future experimental studies are required for a complete characterization of (patho)physiological cholesterol actions [205].

#### 3.3.4. Effects of Gangliosides on RTKs

Gangliosides, the sialic acid-containing glycosphingolipids, can also affect RTKs. Their nomenclature, complex structure, and synthesis pathways are summarized in Figure 4. It has long been shown that GM3 ganglioside strongly inhibits ligand-stimulated autophosphorylation of EGFR and subsequent proliferation in EGFR-overexpressing cancer cells [206]. This was shown to be mediated by carbohydrate-to-carbohydrate interactions between the multiple N-acetylglucosamine termini of the N-glycan moiety of EGFR ECD and the oligosaccharides of GM3 [207]. As suggested by measurements with EGFR reconstituted in proteoliposomes of controlled compositions, this association might consequently result in a block of the ligand-induced allosteric structural transition from inactive to active receptor states through stabilization of the monomer form, which leads to decreased autophosphorylation without affecting ligand binding [208]. Consistently, atomistic MD simulations and their experimental validation using EGFR reconstituted into proteoliposomes confirmed that N-glycosylation of the ECD critically determines the ectodomain orientation relative to the membrane [209]. At the same time, coarse-grained simulations suggested the importance of the K618 residue of the TMD-proximal region of EGFR as a hot spot for highly dynamic interaction with GM3, possibly contributing to the raft-mediated modulation of receptor function [176]. Direct interaction between GM3 and the TMD–JMD segment containing this residue was experimentally confirmed using spectrofluorometric FRET in model membranes [210]. Other RTKs are influenced by membrane gangliosides as well, illustrated by several examples such as GM3-mediated inhibition of FGFR activation and cell proliferation [211], reduction in vascular endothelial growth factor (VEGF)-induced angiogenesis [212], attenuated signaling of insulin receptors contributing to insulin resistance [213], GM1-induced, reduced PDGFR signaling [214], and facilitation of nerve growth factor-stimulated TrkA receptor activation eliciting neurotrophic actions [215].

## 4. Lipid Alterations in Cancer and Possible Applications of Lipid Therapy

Considering the substantial role of various lipids in the functional modulation of membrane proteins, including RTKs, and the importance of these receptors in neoplasms, the connection between lipids and tumors seems obvious. Consistently, cancerous transformations are generally coupled with changes in membrane lipid composition and structure. While the recognition of these alterations is crucial for understanding the connection between tumors, lipids, and RTKs, its detailed analysis is beyond the scope of the current manuscript and, therefore, we limit the discussion to describing the basics in this section and refer to excellent recent reviews in the field.

While oncological research focused mainly on genetic changes and alterations related to proteins in the past, recent advances in lipidomics and membrane biophysics have stimulated studies investigating lipid-related changes and their effects on proteins such as RTKs in tumors [216,217]. Although various normal tissues, and therefore neoplasms originating from them, are characterized by a unique and dynamic set of lipids, altered lipid metabolism and consequent abnormal lipid composition are near-universal hallmarks of human tumors due to increased lipid uptake and reactivation of de novo fatty acid and cholesterol biosynthesis. Aberrant activation of enzymes such as fatty acid synthase, acetyl-CoA carboxylase, ATP-citrate lyase, and transcription factors such as sterol regulatory element binding proteins (SREBPs), liver-X receptors (LXRs), and peroxisome proliferator-activated receptor (PPARs) are typical abnormalities leading to metabolic dysregulation in cancer cells. This dysregulation is a feature of the early stages of tumorigenesis and can be related to elevated growth factor signaling in a feed-forward cycle, as it serves as the key driver of lipid metabolism reprogramming. Through their diverse roles as messengers and regulators of protein function, altered lipid levels can contribute to proliferation, migration, angiogenesis induction, and metastasization of tumor cells and their survival in oxygen- and nutrient-depleted environmental conditions [218,219,220].

### 4.1. Alterations of Phospholipid Levels in Cancer

Cancer cells are generally characterized by substantial changes in the levels of various phospholipids. The neoplastic phenotype is usually associated with elevated levels of these lipids, particularly the most abundant phosphatidylcholine (PC), which is favorable for the higher proliferation rate due to biomass generation for biological membranes and accumulation of highly energetic material. Consistently, alterations in lipid levels were found to correlate with disease progression and patient survival [221,222]. The shift from lipid uptake to de novo lipogenesis in transformed cells leads to increased membrane lipid saturation with higher levels of saturated and monounsaturated chains at the expense of polyunsaturated ones. This results in increased membrane rigidity of tumor cells, contributing to mechanisms of resistance to chemotherapy. Furthermore, reduced levels of polyunsaturated fatty acid chains provide protection from lipid peroxidation [220,223,224]. Besides their levels, the asymmetric distribution of phospholipids in the membrane can also change in tumor cells. For example, PS normally residing in the inner leaflet can be exposed to the outer surface of the cells to induce apoptosis or necroptosis under normal conditions or, in neoplasms, to avoid the recognition as a threat by immune cells and consequent suppression of immune response and thereby promoting tolerance [225]. While the amounts of PC and PS usually increase in cancer cells, phosphatidylethanolamine (PE) levels are often reduced. Besides modulating membrane curvature due to its wedge shape, PE acts as a chaperone assisting protein folding and as a positive regulator of autophagy. Therefore, lack of PE is associated with an excess of unfolded proteins leading to chronic ER stress and accumulation of abnormal cellular organelles promoting the cancerous phenotype [219,226].

Membrane phospholipids can also act as intrinsic elements of signaling pathways in the bilayers. Phosphoinositides (PI), especially their mono, bis, and trisphosphorylated forms (PIP, PIP_2_, and PIP_3_), are important secondary messengers formed in response to growth factor receptor stimulation. Besides their messenger function, PIP molecules are involved in membrane and protein trafficking. At the same time, PIP_2_ acts as a platform for the recruitment of downstream effectors to the membrane, and through electrostatic interactions with proteins, it modifies the functional activity of various ion channels and receptors, including RTKs. PIP_3_ is formed from PIP_2_ by PI3K and activates Akt as a central mediator of the PI3K/Akt/mTORC1 pathway commonly involved in the pathogenesis of various tumors through promoting cell growth and survival [226,227].

### 4.2. Alterations of (Glycol)Sphingolipid Levels in Cancer

Sphingolipids represent another essential group of bioactive lipids. The two most widely studied members of this lipid group are ceramide involved in many cellular stress responses, including the regulation of apoptosis and cell senescence, and sphingosine-1-phosphate (S1P), which plays important roles in cell survival, migration, and inflammation in a dichotomous way. Their levels are regulated by a highly complex interconnected network of enzymes generating a great variety of sphingolipids and their specific and varied trafficking and subcellular localization. S1P, a sphingolipid having sufficient aqueous solubility to act as a soluble mediator, is typically generated from sphingosine via sphingosine kinase in response to growth factor or cytokine signals. It can be considered as a trace lipid due to its very low physiological concentration and, therefore, it exerts its action via binding to the S1P receptor, a high-affinity G protein-coupled receptor. S1P signaling leads to cell survival, proliferation, migration, cell growth, and chemotherapeutic resistance, and, therefore, its level is usually elevated in cancer cells, typically through overexpression of sphingosine kinase 1 or 2. On the other hand, the de novo pathway of sphingolipid synthesis and activation of acid or neutral sphingomyelinases leading to ceramide generation are typically activated in response to stress signals such as chemotherapeutic agents, death receptors, or ionizing radiation. While the highly hydrophobic ceramides can also affect proteins through direct binding, such as in ceramide-activated Ser–Thr phosphatases (CAPPs) PP1 and PP2A or lysosomal cathepsin D, they can form ceramide platforms, very rigid membrane microdomains recruiting signaling molecules to initiate pathways leading to apoptosis, necroptosis, or lethal mitophagy—an antitumorigenic macroautophagy mechanism resulting in degradation of mitochondria and eventual cell death. Consequently, ceramide levels are commonly decreased in cancer cells, often due to changes in the expression levels of ceramide synthase enzymes generating ceramides of different lengths [228,229,230].

The presence of oligosaccharides or hydroxylated fatty acids is unique to sphingolipids. Sphingolipids with 2-hydroxy fatty acid as the N-acyl chain of ceramide are generated by fatty acid 2-hydroxylase, an enzyme often upregulated in cancer [231,232]. While the exact way by which accumulation of hydroxylated lipids contributes to carcinogenesis remains to be elucidated, from the standpoint of membrane biophysics, hydroxylated fatty acids have been shown to increase membrane rigidity by participating in an extensive network of hydrogen bonds [231]. Gangliosides, sialic acid-containing glycosphingolipids, were also shown to be involved in tumorigenesis. While disialogangliosides generally enhance cancerous phenotypes by immunosuppression, angiogenesis, and regulation of proliferation, cell adhesion, motility, and metastasis, monosialogangliosides usually suppress them. Neoexpression of disialogangliosides such as GD2 and GD3, and altered expression of glycosyltransferase enzymes involved in their synthesis, can be observed in several neuro-ectoderm-derived tumors. Their increased levels are strongly associated with the metastatic potential, invasiveness of cancer cells, and poor prognosis. Resulting from the crucial roles of GD3 and GD2 in melanoma and small-cell lung cancer, respectively, these gangliosides were identified as disease-associated biomarkers. Furthermore, certain gangliosides, including 9-O-acetyl-GD3, 9-O-acetyl-GT3, or N-glycolyl-GM3, which are normally missing or expressed at very low levels, were shown to be elevated in breast cancer. Similarly, fucosyl-GM1 expression can be considered as a characteristic feature of small-cell lung cancers. Oppositely, the monosialoganglioside GM3 generally inhibits cell proliferation, and angiogenesis and tumors expressing high levels of this ganglioside are characterized by slower growth [233,234].

### 4.3. Alterations of Cholesterol Levels in Cancer

Increased cholesterol levels belong to the hallmarks of cancer cell metabolism, which can result from oncogene signaling-induced excess uptake due to higher LDL receptor and NPC1L1 levels, dysregulation of cellular efflux through ABCA1, and upregulation of the master transcription factor SREBP2 and its downstream targets, such as HMG-CoA reductase, squalene synthase, squalene epoxidase, and oxidosqualene cyclase, the rate-limiting enzymes of cholesterol biosynthesis. Furthermore, activation of this pathway leads to extensive protein prenylation participating in proliferative, migratory, and survival signaling mechanisms and production of ubiquinone, contributing to protection from lipid peroxidation, which promotes tumorigenesis. Cholesterol enrichment in tumors is generally associated with upregulation of acyl-CoA cholesterol acyltransferase (ACAT1) and its consequent strongly oncogenic storage as lipid droplets in the forms of cholesteryl esters. These droplets can serve as reservoirs available upon increased demand. In cancer cells, cholesterol can be converted into oxysterols, signaling mediators substantially affecting the tumor microenvironment through recruitment of immunosuppressive neutrophils and tumor-associated macrophages with an M2-phenotype and inhibition of immune-effector T cells and antigen presentation of dendritic cells, which facilitate the escape of tumors from immune surveillance [235,236,237]. Furthermore, besides affecting functional activities of proteins involved in the proliferation signaling cascades via membrane biophysical alterations and lipid raft-mediated effects, cholesterol was shown to exert its action via direct binding. An increased amount of cholesterol was shown to promote epithelial–mesenchymal transition—a tumor cell plasticity program resulting in repression of the epithelial and activation of the mesenchymal phenotype—facilitating a hybrid phenotype with the acquisition of survival programs, sustained resistance to chemotherapeutics, and elevated metastatic potential. These actions are mediated by raft recruitment of elements involved in Wnt and transforming growth factor β (TGFβ) signaling pathways and direct actions on the sterol sensing domains of Patched, the receptor of Sonic hedgehog ligand, and Smoothened, a G protein-coupled receptor serving as an effector molecule. Therefore, cholesterol substantially influences the three major pathways of epithelial–mesenchymal transition [109,238].

In tumors, elevated cholesterol levels have been described in mitochondria as well, primarily due to the steroidogenic acute regulatory domain 1 (StARD1), a transport protein in the mitochondrial outer membrane. Increased amounts of cholesterol were suggested to contribute to the Warburg effect, i.e., dependence on anaerobic glycolysis despite normal oxygen tension, which ensures ATP generation at low oxygen levels, and, moreover, the production of metabolites favoring unrestrained growth and proliferation, invasion, angiogenesis, and immunosurveillance suppression. These effects participating in the orchestration of metabolic reprogramming were proposed to be mediated through increased mitochondrial ROS levels and stabilization of hypoxia-inducible factor-1α. Furthermore, mitochondrial cholesterol is involved in resistance to BAX-mediated apoptosis induced by chemotherapeutic agents eliciting mitochondrial outer membrane permeabilization [237,239].

The elevated levels of cholesterol and sphingolipids, and consequently increased membrane thickness and rigidity, substantially contribute to resistance to chemotherapy by various mechanisms, i.e., through reduced drug influx resulting from increased membrane lipid density, drug entrapment, and subsequent exocytosis by altering fluidity-dependent endocytic and sorting mechanisms, or increased drug efflux by facilitating the drug binding and pumping activities of P-glycoproteins and other multidrug resistance-related ABC transporters through specific binding or modulation of hydrophobic matching and lipid raft-partitioning [217,238,240].

### 4.4. Relevance of Lipid Alterations in Tumor Diagnosis and Therapy

Due to the magnitude and specific feature of alterations in lipid levels and technological advances enabling simultaneous, quantitative analysis of a large number of different lipid species, characteristic changes in lipid profiles can be utilized as potential biomarkers for certain cancer types. Furthermore, these specific lipid signatures can be used to follow disease state and therapeutic response [220,233,241].

Considering the crucial importance of lipid alterations in neoplasms, a relatively novel therapeutic concept, membrane lipid therapy aimed at modifying membrane lipid composition and structure, is emerging as a promising, efficient, and specific alternative to conventional chemotherapies. A huge variety of lipid therapy approaches were described in the literature, demonstrating their beneficial effects when applied alone or in combination with standard chemotherapeutics in preclinical studies and clinical trials as well. These methods can be classified based on their mechanisms of action, as described below with several examples.

(1) Regulation of the activity of enzymes involved in the biosynthesis and metabolism of membrane lipids: inhibition of phospholipid synthesis by fatty acid synthase inhibitor (orlistat), inhibition of cholesterol synthesis by blocking HMG-CoA reductase (statins), oxidosqualene cyclase (Ro 48-8071), and squalene synthase (zaragozic acid), decreasing cholesterol esterification and storage to induce apoptosis due to an overload of free cholesterol with ACAT1 inhibitors (CP-113818, bitter melon extract, avasimin (a nanoformulation containing avasimibe)), increasing ceramide levels through ceramidase inhibition (B13, LCL-464, and KPB-27), or lowering S1P levels by blocking sphingosine kinase 1 (N,N-dimethylsphingosine) [219,233,236,242,243].

(2) Modulation of gene expression that results in changes in membrane lipid composition: inhibition of SREBPs (fatostatin), activation of PPARα (fenofibrate), or LXR (agonists T0901317, GW3965, and LXR623 or inverse agonist SR9243) or decreasing cholesterol uptake with LDLR silencing [219,233,236,242,243].

(3) Modification of bulk biophysical parameters and lateral microdomain organization of the cell membrane: disruption or modification of lipid raft structure (statins, various cyclodextrin derivatives or dietary fish oil with polyunsaturated fatty acids, especially docosahexaenoic acid), induction of apoptosis through promoting clustering of death receptors via ceramide platforms (short-chain ceramides, synthetic alkylphospholipids such as edelfosine, miltefosine and perifosine and plant-derived compounds such as activin D or resveratrol) or modification of bulk lipid composition (2-hydroxyoleic acid causing order reduction by activating SM synthase leading to SM increase and PE decrease) [219,233,236,242,243].

(4) Approaches targeting cancer-associated molecules: These methods mainly include immunotherapeutic approaches, e.g., antibodies against GD3 (anti-GD3 R24), GD2 (Dintuximab, anti-GD2 14G2a), fucosyl-GM1 (BMS-986012), PS (bavituximab), and S1P (neutralizing antibody sphingomab) and chimeric antigen receptor T-cell therapy in GD2-positive solid tumors [219,233,236,242,243]. Due to the preferential presence of hydroxylated lipids, cancer cells can be targeted by a marine-derived cyclic depsipeptide, elisidepsin [244]. The widespread hypoxia in advanced tumors, leading to inhibition of oxygen-requiring reactions, including hydroxylation, may explain the lack of antitumor effect of elisidepsin in advanced cancer [245,246].

## 5. Conclusions

According to Max Planck, “Insight must precede application”. The history of how insight into the workings of RTKs has been gained, followed by the rational application of small-molecule tyrosine kinase inhibitors and antireceptor antibodies, attests to the validity of the quotation. The limited potential of these targeted therapies calls for further innovations in the field [247]. This review was aimed at summarizing recent insight into the intricate details of how different kinds and levels of receptor clusters and their interactions with the plasma membrane determine signaling. Receptor activation is determined by the balance of factors promoting and inhibiting the formation of active receptor dimers or higher-order oligomers. This principle, schematically summarized in Figure 5, provides ample opportunities for the development of therapeutic approaches with fundamentally novel targets.

## Figures and Tables

**Figure 1 cancers-14-00944-f001:**
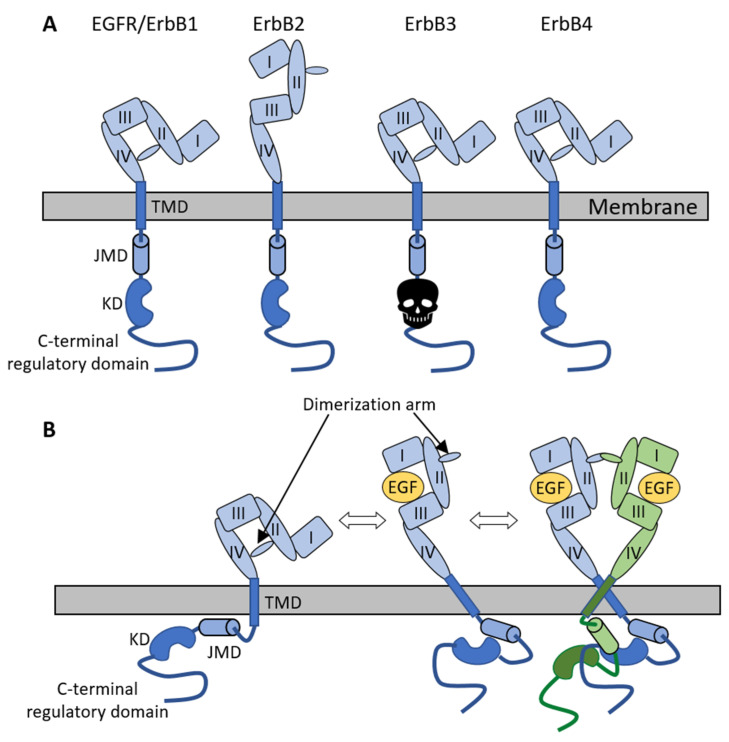
Domain organization and activation of ErbB receptors. (**A**) All four members of the ErbB family follow the same domain organization. Panel A displays the receptors in the absence of ligand. The extracellular domain (ECD) adopts a closed conformation in EGFR, ErbB3, and ErbB4. Although the ECD of ErbB2 assumes an extended structure with the dimerization arm exposed, several autoinhibitory interactions have been identified in it (not shown in the figure). The ligand-binding pocket of ErbB2 is constrained, preventing the binding of growth factors. The extracellular portion is followed by the transmembrane domain (TMD), and by the juxtamembrane (JMD), the kinase (KD), and the C-terminal regulatory domains on the intracellular side. Due to mutations in key amino acid residues, the KD of ErbB3 displays reduced or no activity at all. (**B**) The monomeric EGFR on the left displays the structure of the receptor in the absence of ligand. The ECD adopts a closed conformation in which the dimerization arm is buried inside the protein forming an intramolecular bridge between domains II and IV. Upon ligand binding to domains I and III, the ECD undergoes a large rearrangement resulting in the formation of the ligand-binding pocket and exposure of the dimerization arm that is now ready to form an intermolecular bridge with another EGFR whose extracellular domain also adopts this extended conformation (dimeric structure on the right). In the active receptor dimer, the TMDs form a dimer stabilized by the N-terminal dimerization motif, the JMD dissociates from anionic phospholipids of the membrane, and the KDs form an asymmetric dimer.

**Figure 2 cancers-14-00944-f002:**
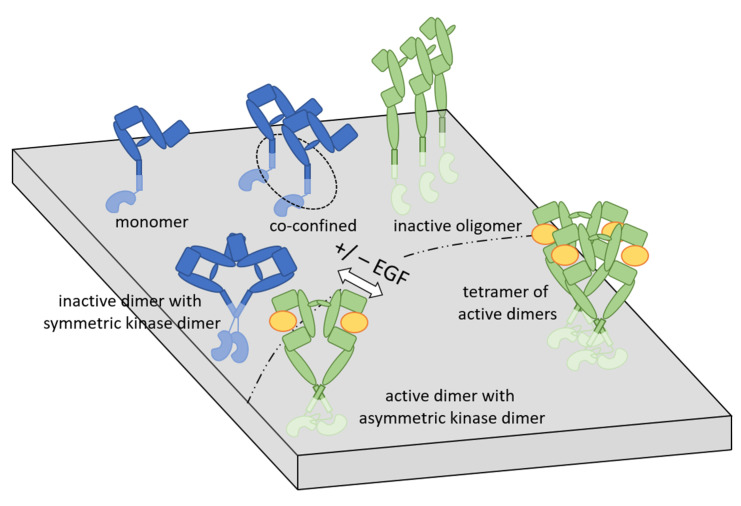
Model of how clustering contributes to the regulation of EGFR. In the absence of EGF (orange oval), the extracellular part of the EGFR adopts a closed conformation (blue-colored objects) and is either monomeric or forms different kinds of preformed clusters. Some of these associations arise as a result of the co-confinement of monomeric receptors in the vicinity of each other. The receptor proteins in these loose associations are unlikely to interact with each other, but this cluster type predisposes them to form direct molecular associations. Unliganded EGFRs with their ECD in the closed conformation can form dimers in which the C-terminal dimerization motif of the TMD (close to the intracellular leaflet of the plasma membrane) interact, and the kinases form an inactive, symmetric dimer. Ligandless EGFR can also undergo oligomerization with the ECD in the extended conformation (displayed in green), and the KDs are positioned in a way that prevents their interaction. Upon EGF challenge, the extended conformation of the ECD is stabilized, and the exposed dimerization arms lead to the formation of back-to-back dimers of the ECD, which brings the N-terminal dimerization motifs of the TMD into close proximity and results in the formation of asymmetric KD dimers capable of activation. Such back-to-back dimers most likely undergo oligomerization in order to enable phosphorylation of the C-terminal regulatory domains (the C-terminal regulatory domains are not shown in the figure).

**Figure 3 cancers-14-00944-f003:**
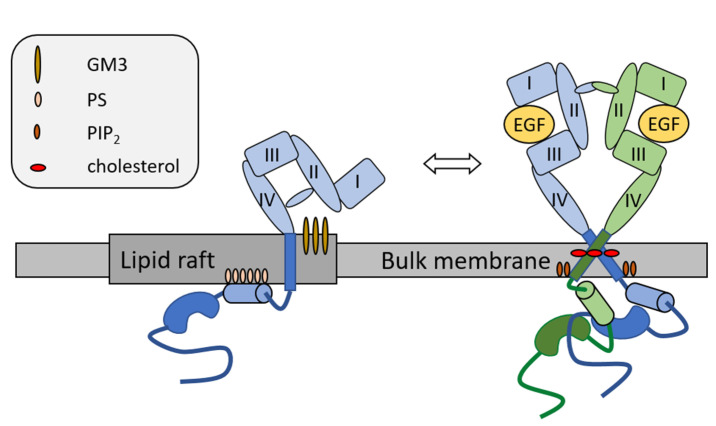
A brief overview of lipid-mediated effects on the EGF receptor. The inactive monomeric form of the receptor, displayed on the left, is stabilized by (i) direct interactions between multiple N-acetylglucosamine termini of the N-glycan moiety in the EGFR ECD and the oligosaccharides of GM3, and (ii) by electrostatic interactions between anionic PS species of the inner leaflet and three positively charged amino acid clusters of the JM-A segment resulting in membrane embedding of the JMD. Ligand binding induces conformational changes leading to the formation of the active dimer configuration, which involves cholesterol-mediated dimerization of the N-terminal GxxxG motifs of the TMDs. It is also facilitated by direct electrostatic interactions between PIP_2_ lipids of the inner leaflet and proximal basic residues of the JM-A, helping the JMD being released from the membrane and formation of the antiparallel JMD dimer and the active, asymmetric kinase dimer. The process is accompanied by the exit of the EGFR from the lipid raft microdomains to the bulk membrane phase, releasing the receptors from inhibitory constraints.

**Figure 4 cancers-14-00944-f004:**
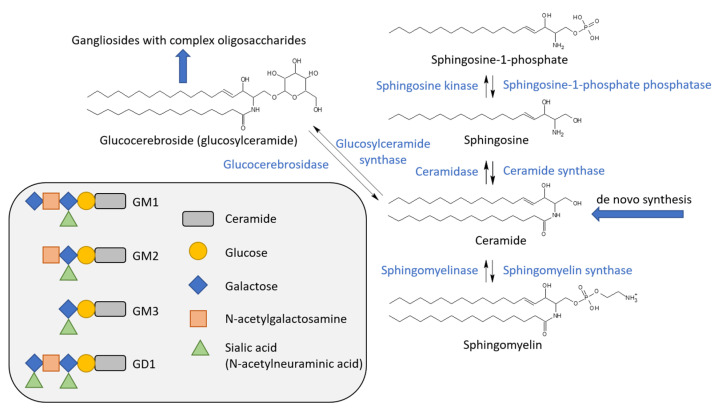
Brief biochemistry of sphingolipids. Ceramide is at the center of the biochemical pathways of sphingolipids. Enzymes are written in blue in the figure. De novo synthesis of ceramide from palmitate and serine takes place in the endoplasmic reticulum. It can be converted to sphingomyelin by the transfer of phosphocholine from phosphatidylcholine by sphingomyelin synthase located in the Golgi complex or at the plasma membrane. By cleaving the fatty acid from ceramide, ceramidase produces sphingosine that can be further phosphorylated to sphingosine-1-phosphate. Addition of glucose to ceramide results in glucocerebroside from which gangliosides with complex oligosaccharides can be synthesized. The nomenclature of gangliosides is briefly summarized in the insert on the left.

**Figure 5 cancers-14-00944-f005:**
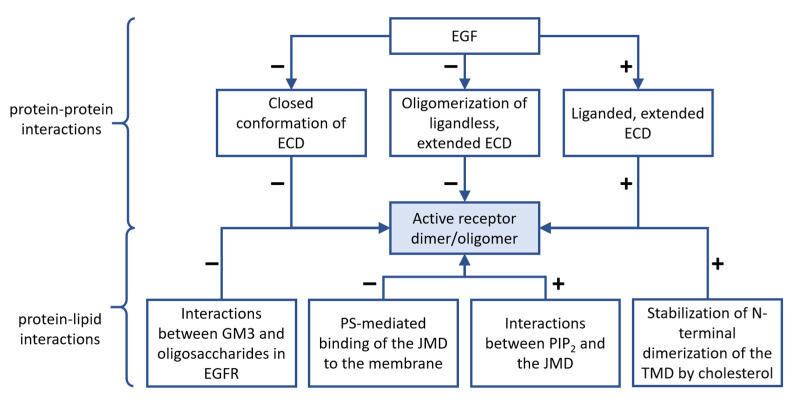
The formation of the EGFR signaling unit is at the crossroads of activating and inhibitory factors. The physiological aim of the complex network governing receptor dimerization and oligomerization is to keep ligandless receptors in an inactive state and to enable the formation of signaling competent units after growth factor binding. Ligandless ECDs prevent the formation of active, signaling competent dimers/oligomers, while the ligand-bound ECD favors their formation by tilting the TMDs such that they favor the formation of asymmetric KD dimers (not explicitly shown in the figure). These processes involve conformational transitions of the receptors themselves, but they are supported by protein–lipid interactions that can also foster or block the formation of active receptors, which must probably form larger-order associations to become signaling competent.

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
