# Peer review of "It Takes More than Two to Tango: Complex, Hierarchal, and Membrane-Modulated Interactions in the Regulation of Receptor Tyrosine Kinases"

_cancers, 2022, doi:10.3390/cancers14040944_

Round 1
Reviewer 1 Report
This paper provides a concise overview about important subfamily of transmembrane receptors which are Receptor tyrosine kinases and their role in regulating the balance between cell proliferation and cell death.
It would be interesting for the authors to present all or most of the topics discussed on an additional figure. The authors are able to prepare such an additional Figure?
Author Response
We thank the reviewer for the positive overall comments. Below please find our response to the criticism raised in the review.
It would be interesting for the authors to present all or most of the topics discussed on an additional figure. The authors are able to prepare such an additional Figure?
Response: We have added a new figure to the Conclusion part of the manuscript describing the balance between factors promoting and inhibiting the formation of active signaling units.
Reviewer 2 Report
This is a comprehensive review of the regulation of receptor tyrosine kinases and then pivots towards discussing lipid alterations in cancer and the possibility of modifying lipid membrane compositions as an alternative cancer therapy. The review contains 13 sections, including an introductory and a discussion section. But, having just five sections and their corresponding subsections would improve the organization of this review: Being section 1, introduction; section 2, dimerization-induced activation of receptor tyrosine kinase (with four subsections); section 3, Protein-lipid interaction and their examination (with three subsections); section 4, Lipid alterations in cancer and possible applications of lipid therapy (with four subsections).
Also, reading the simple summary, the abstract, and the introduction, one has the impression that the whole review is about RTK regulation and its role in cancer development and progression. But, the second part of this review is focused on the association between lipid alteration and cancer cell regulation. This last part is not related to the regulation of RTK, so the reviewer suggests including this information in the abstract, introduction, and title and making a better transition between these two parts.
Minor comments:
- Figure 1 would benefit if the monomeric structure together with the ligand is included in the cartoon, along with a label of the dimerization arm. Also, It is not clear whether the ligand binds to only domains I and II or also binds to domain III.
- A cartoon representing the ErbB1, 2, 3, and 4 structures will help to visualize the differences between them.
Author Response
We are thankful to the reviewer for the positive comments and for suggesting ways to improve the manuscript. Below please find out detailed responses to the criticism.
The review contains 13 sections, including an introductory and a discussion section. But, having just five sections and their corresponding subsections would improve the organization of this review…
Response: We have restructured the manuscript according to the suggestion of the reviewer.
Also, reading the simple summary, the abstract, and the introduction, one has the impression that the whole review is about RTK regulation and its role in cancer development and progression. But, the second part of this review is focused on the association between lipid alteration and cancer cell regulation. This last part is not related to the regulation of RTK, so the reviewer suggests including this information in the abstract, introduction, and title and making a better transition between these two parts.
Response: The simple summary, the abstract and the introduction have been modified to reflect the fact that a significant part of the paper is devoted to protein-lipid interactions and their potential role in cancer. Furthermore, section 3 devoted to protein-lipid interactions has been supplemented with an introductory paragraph making the transition between the two major parts of the manuscript smoother.
Minor comments:
Figure 1 would benefit if the monomeric structure together with the ligand is included in the cartoon, along with a label of the dimerization arm. Also, It is not clear whether the ligand binds to only domains I and II or also binds to domain III.
Response: Figure 1 in the revised manuscript displays the monomeric, liganded receptor. Furthermore, the ligand is now shown to bind domains I and III only.
A cartoon representing the ErbB1, 2, 3, and 4 structures will help to visualize the differences between them.
Response: The updated version of Figure 1 contains a comparison of all ErbB structures.